# Power and limitations of single-qubit native quantum neural networks

**Zhan Yu**         **Hongshun Yao**         **Mujin Li**

**Xin Wang**[*]
Institute for Quantum Computing, Baidu Research, Beijing 100193, China[†]

## Abstract

Quantum neural networks (QNNs) have emerged as a leading strategy to establish applications in machine learning, chemistry, and optimization. While the applications of QNN have been widely investigated, its theoretical foundation remains less understood. In this paper, we formulate a theoretical framework for the expressive ability of data re-uploading quantum neural networks that consist of interleaved encoding circuit blocks and trainable circuit blocks. First, we prove that single-qubit quantum neural networks can approximate any univariate function by mapping the model to a partial Fourier series. We in particular establish the exact correlations between the parameters of the trainable gates and the Fourier coefficients, resolving an open problem on the universal approximation property of QNN. Second, we discuss the limitations of single-qubit native QNNs on approximating multivariate functions by analyzing the frequency spectrum and the flexibility of Fourier coefficients. We further demonstrate the expressivity and limitations of single-qubit native QNNs via numerical experiments. We believe these results would improve our understanding of QNNs and provide a helpful guideline for designing powerful QNNs for machine learning tasks.

## 1  Introduction

Quantum computing is a technology that exploits the laws of quantum mechanics to solve complicated problems much faster than classical computers. It has been applied in areas such as breaking cryptographic systems [1], searching databases [2], and quantum simulation [3, 4], in which it gives a quantum speedup over the best known classical algorithms. With the fast development of quantum hardware, recent results [5–7] have shown quantum advantages in specific tasks. An emerging direction is to investigate if quantum computing can offer quantum advantages in artificial intelligence, giving rise to an interdisciplinary area called *quantum machine learning* [8].

A leading strategy to quantum machine learning uses *quantum neural networks* (QNNs), which are quantum analogs of artificial neural networks (NNs). Much progress has been made in applications of QNN in various topics [9–11], including quantum autoencoder [12, 13], supervised learning [14–17], dynamic learning [18–20], quantum chemistry [21], and quantum metrology [22–24]. Similar to the field of machine learning, a crucial challenge of quantum machine learning is to design powerful and efficient QNN models for quantum learning tasks, which requires a theoretical understanding of how structural properties of QNN may affect its expressive power.

The expressive power of a QNN model can be characterized by the function classes that it can approximate. Recently, the universal approximation property (UAP) of QNN models has been

---

[*]Corresponding author. wangxin73@baidu.com
[†]Z. Y. and H. Y. contributed equally to this work.

36th Conference on Neural Information Processing Systems (NeurIPS 2022).

investigated, which is similar to the universal approximation theorem [25, 26] in machine learning theory. The authors of [27] suggested that a QNN model can be written as a partial Fourier series in the data and proved the existence of a multi-qubit QNN model that can realize a universal function approximator. The UAP of single-qubit models remains an open conjecture, due to the difficulties in analyzing the flexibility of Fourier coefficients. Another work [28] considered hybrid classical-quantum neural networks and obtained the UAP by using the Stone-Weierstrass theorem. Ref. [29] proved that even a single-qubit hybrid QNN can approximate any bounded function.

The above results of UAP show that the expressivity of QNNs is strong, but it does not reveal the relationship between the structural properties of a QNN and its expressive ability. Therefore the UAP may not be a good guide for constructing QNN models with practical interests. In particular, it is worth noting that the existence proof in Ref. [27] is under the assumption of multi-qubit systems, exponential circuit depth, and arbitrary observables, which does not explicitly give the structure of QNNs. Meanwhile, Refs. [28, 29] demonstrated the construction of QNNs in detail, but it is unclear whether the powerful expressivity comes from the classical part or the quantum part of hybrid models. Moreover, a systematic analysis of how parameters in the QNN affect the classes of functions that it can approximate is missing. The absence of these theoretical foundations hinders the understanding on the expressive power and limitation of QNNs, which makes it highly necessary but challenging to design effective and efficient QNNs.

To theoretically investigate the expressivity of QNNs, it is important to study the simplest case of single-qubit QNNs, just like the celebrated universal approximation theorem first showing the expressivity of depth-2 NNs [25, 26]. In this paper, we formulate an analytical framework that correlates the structural properties of a single-qubit native QNN and its expressive power. We consider data re-uploading models that consist of interleaved data encoding circuit blocks and trainable circuit blocks [30]. First, we prove that there exists a single-qubit native QNN that can express any Fourier series, which is a universal approximator for any square-integrable univariate function. It solves the open problem on the UAP of single-qubit QNNs in Ref. [27]. Second, we systematically analyze how parameters in trainable circuit blocks affect the Fourier coefficients. The main results on the expressivity of QNNs are summarized as in Fig. 1. Third, we discuss potential difficulties for single-qubit native QNNs to approximate multivariate functions. Additionally, we compare native QNNs with the hybrid version and show the fundamental difference in their expressive power. We also demonstrate the expressivity and limitations of single-qubit native QNNs via numerical experiments on approximating univariate and multivariate functions. Our analysis, beyond the UAP of QNNs, improves the understanding of the relationship between the expressive power and the structure of QNNs. This fundamental framework provides a theoretical foundation for data re-uploading QNN models, which is helpful to construct effective and efficient QNNs for quantum machine learning tasks.

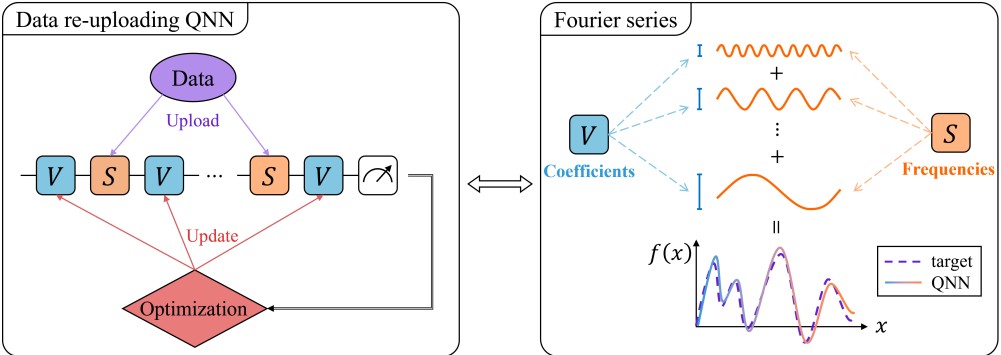

Figure 1: **A schematic diagram of the main results.** A single-qubit data re-uploading QNN model consisting of interleaved trainable blocks (in blue) and encoding blocks (in orange) corresponds to a partial Fourier series. The encoding blocks decide the frequency spectrum of the Fourier series and the trainable blocks control the Fourier coefficients. By the summability and convergence of the Fourier series, the QNN model can approximate any square-integrable target function.

We will start by giving some background and defining the native QNN models in the next section, and then analyze the expressivity of single-qubit native QNNs in Section 3. In Section 4, we discuss

the limitation of single-qubit native QNNs and compare native QNNs with hybrid QNNs, which shows the fundamental difference between their expressive power. The numerical experiments on the expressivity and limitations of single-qubit native QNNs are described in Section 5.

## 2 Preliminaries

### 2.1 A primer on quantum computing

**Quantum state** The basic unit of information in quantum computation is one *quantum bit*, or *qubit* for short. Just like a classical bit has a state in either 0 or 1, a qubit also has a state. A single-qubit state is a unit vector in a 2-dimensional Hilbert space $\mathbb{C}^2$, which is commonly denoted in Dirac notation $|\psi\rangle = \alpha |0\rangle + \beta |1\rangle$, where $|0\rangle = (1,0)^T$ and $|1\rangle = (0,1)^T$ are known as computational basis states. Here $|\psi\rangle$ denotes a column vector and its conjugate transpose $\langle\psi| := |\psi\rangle^\dagger$ is a row vector. Then the inner product $\langle\psi|\psi\rangle = \|\psi\|^2$ denotes the square of $L^2$-norm of $|\psi\rangle$. Note that $|\psi\rangle$ is a normalized state so $\langle\psi|\psi\rangle = |\alpha|^2 + |\beta|^2 = 1$. Having this constraint, a single-qubit state can be represented as a point at surface of a *Bloch sphere*, written as $|\psi\rangle = \cos(\theta/2) |0\rangle + e^{i\phi} \sin(\theta/2) |1\rangle$, where $\theta$ and $\phi$ are re-interpreted as azimuthal angle and polar angle in spherical coordinates. More generally, a quantum state of $n$ qubits can be represented as a normalized vector in the $n$-fold tensor product Hilbert space $\mathbb{C}^{2^n}$.

**Quantum gate** Quantum gates are basic operations used to manipulate qubits. Unlike some classical logical gates, quantum gates are reversible, so they can be represented as *unitary transformations* in the Hilbert space. A unitary matrix $U$ satisfies $U^\dagger U = UU^\dagger = I$. A commonly used group of single-qubit quantum gates is the *Pauli gates*, which can be written as Pauli matrices:

$$X = \begin{bmatrix} 0 & 1 \\ 1 & 0 \end{bmatrix}, \qquad Y = \begin{bmatrix} 0 & -i \\ i & 0 \end{bmatrix}, \qquad Z = \begin{bmatrix} 1 & 0 \\ 0 & -1 \end{bmatrix}. \tag{1}$$

The Pauli $X$, $Y$, and $Z$ gates are equivalent to a rotation around the $x$, $y$, and $z$ axes of the Bloch sphere by $\pi$ radians, respectively. A group of more general gates is the *rotation operator gates* $\{R_P(\theta) = e^{-i\frac{\theta}{2}P} \mid P \in \{X, Y, Z\}\}$, which allows the rotating angle around the $x$, $y$ and $z$ axes of the Bloch sphere to be customized. They can be written in the matrix form as

$$R_X(\theta) = \begin{bmatrix} \cos\frac{\theta}{2} & -i\sin\frac{\theta}{2} \\ -i\sin\frac{\theta}{2} & \cos\frac{\theta}{2} \end{bmatrix}, \quad R_Y(\theta) = \begin{bmatrix} \cos\frac{\theta}{2} & -\sin\frac{\theta}{2} \\ \sin\frac{\theta}{2} & \cos\frac{\theta}{2} \end{bmatrix}, \quad R_Z(\theta) = \begin{bmatrix} e^{-i\frac{\theta}{2}} & 0 \\ 0 & e^{i\frac{\theta}{2}} \end{bmatrix}. \tag{2}$$

**Quantum measurement** A measurement is a quantum operation to retrieve classical information from a quantum state. The simplest measurement is the computational basis measurement; for a single-qubit state $|\psi\rangle = \alpha |0\rangle + \beta |1\rangle$, the outcome of such a measurement is either $|0\rangle$ with probability $|\alpha|^2$ or $|1\rangle$ with probability $|\beta|^2$. Computational basis measurements can be generalized to *Pauli measurements*, where Pauli matrices are *observables* that we can measure. For example, measuring Pauli $Z$ is equivalent to the computational basis measurement, since $|0\rangle$ and $|1\rangle$ are eigenvectors of $Z$ with corresponding eigenvalues $\pm 1$. Pauli $Z$ measurement returns $+1$ if the resulting state is $|0\rangle$ and returns $-1$ if the resulting state is $|1\rangle$. We can calculate the expected value of Pauli $Z$ measurement when the state is $|\psi\rangle$:

$$\langle\psi| Z |\psi\rangle = (\alpha^* \langle 0| + \beta^* \langle 1|)Z(\alpha |0\rangle + \beta |1\rangle) = |\alpha|^2 - |\beta|^2. \tag{3}$$

Pauli measurements can be extended to the case of multiple qubits by a tensor product of Pauli matrices.

### 2.2 Data re-uploading quantum neural networks

We consider the data re-uploading QNN model [30], which is a generalized framework of quantum machine learning models based on parameterized quantum circuits [31]. A data re-uploading QNN is a quantum circuit that consists of interleaved data encoding circuit blocks $S(\cdot)$ and trainable circuit blocks $V(\cdot)$,

$$U_{\boldsymbol{\theta},L}(\boldsymbol{x}) = V(\boldsymbol{\theta_0}) \prod_{j=1}^{L} S(\boldsymbol{x})V(\boldsymbol{\theta_j}), \tag{4}$$

where $x$ is the input data, $\boldsymbol{\theta} = (\boldsymbol{\theta_0}, \ldots, \boldsymbol{\theta_L})$ is a set of trainable parameters, and $L$ denotes the number of layers. It is common to build the data encoding blocks and trainable blocks using the most prevalent parameterized quantum operators $\{R_X, R_Y, R_Z\}$. We define the output of this model as the expectation value of measuring some observable $M$,

$$f_{\boldsymbol{\theta},L}(\boldsymbol{x}) = \langle 0 | \, U_{\boldsymbol{\theta},L}^{\dagger}(\boldsymbol{x}) M U_{\boldsymbol{\theta},L}(\boldsymbol{x}) \, | 0 \rangle \, . \tag{5}$$

Note that some data re-uploading QNNs introduce trainable weights in data pre-processing or post-processing, which are considered as hybrid QNNs. For example, the data encoding block defined as $S(\boldsymbol{w} \cdot \boldsymbol{x})$ is essentially equivalent to feeding data $\boldsymbol{x}$ into a neuron with weight $\boldsymbol{w}$ and then uploading the output to an encoding block $S(\cdot)$. Such a mixing structure makes it hard to tell whether the expressive power comes from the classical or quantum part. To solely study the expressive power of QNNs, we define the concept of *native QNN*, where all trainable weights are introduced by parameters of tunable quantum gates so that they can be distinguished from a hybrid QNN. Throughout this paper, we simply refer to the native QNN as QNN for short unless specified otherwise.

## 3 Expressivity of single-qubit QNNs

To better understand the expressive power of QNNs, we start investigating the simplest case of single-qubit models. Ref. [27] investigated the expressive power of QNNs using the Fourier series formalism. In this section, we establish an exact correlation between the single-qubit QNN and the Fourier series in terms of both the frequency spectrum and Fourier coefficients. Note that we consider one-dimensional input data for now, which corresponds to the class of univariate functions.

A Fourier series is an expansion of a periodic function $f(x)$ in infinite terms of a sum of sines and cosines which can be written in the exponential form as

$$f(x) = \sum_{n=-\infty}^{\infty} c_n e^{i \frac{2\pi}{T} n x}, \tag{6}$$

where

$$c_n = \frac{1}{T} \int_T f(x) e^{i \frac{2\pi}{T} n x} dx \tag{7}$$

are the Fourier coefficients. Here $T$ is the period of function $f(x)$. The quantities $n\frac{2\pi}{T}$ are called the *frequencies*, which are multiples of the base frequency $\frac{2\pi}{T}$. The set of frequency $\{n\frac{2\pi}{T}\}_n$ is called the *frequency spectrum* of Fourier series.

In approximation theory, a partial Fourier series (or truncated Fourier series)

$$s_N(x) = \sum_{n=-N}^{N} c_n e^{i \frac{\pi}{T} n x} \tag{8}$$

is commonly used to approximate the function $f(x)$. A partial Fourier series can be transformed to a Laurent polynomial $P \in \mathbb{C}[z, z^{-1}]$ by the substitution $z = e^{i \frac{2\pi}{T} x}$, i.e.,

$$P(z) = \sum_{n=-N}^{N} c_n z^n. \tag{9}$$

A Laurent polynomial $P \in \mathbb{F}[z, z^{-1}]$ is a linear combination of positive and negative powers of the variable $z$ with coefficients in $\mathbb{F}$. The *degree* of a Laurent polynomial $P$ is the maximum absolute value of any exponent of $z$ with non-zero coefficients, denoted by $\deg(P)$. We say that a Laurent polynomial $P$ has *parity* 0 if all coefficients corresponding to odd powers of $z$ are 0, and similarly $P$ has parity 1 if all coefficients corresponding to even powers of $z$ are 0.

Following the pattern of Fourier series, we first consider using $R_Z(x) = e^{-ixZ/2}$ to encode the input $x$ and let $R_Y(\cdot)$ be the trainable gate. We can write the QNN as

$$U_{\boldsymbol{\theta},L}^{YZY}(x) = R_Y(\theta_0) \prod_{j=1}^{L} R_Z(x) R_Y(\theta_j), \tag{10}$$

and the quantum circuit is shown in Fig. 2.

Figure 2: Circuit of $U_{\theta,L}^{YZY}(x)$, where the trainable block is $R_Y(\cdot)$ and the encoding block is $R_Z(\cdot)$.

To characterize the expressivity of this kind of basic QNN, we first rigorously show that the QNN $U_{\theta,L}^{YZY}(x)$ can be represented in the form of a partial Fourier series with real coefficients.

**Lemma 1** *There exist $\boldsymbol{\theta} = (\theta_0, \theta_1, \ldots, \theta_L) \in \mathbb{R}^{L+1}$ such that*

$$U_{\theta,L}^{YZY}(x) = \begin{bmatrix} P(x) & -Q(x) \\ Q^*(x) & P^*(x) \end{bmatrix} \tag{11}$$

*if and only if real Laurent polynomials $P, Q \in \mathbb{R}[e^{ix/2}, e^{-ix/2}]$ satisfy*

1. $\deg(P) \le L$ *and* $\deg(Q) \le L$,

2. $P$ *and* $Q$ *have parity* $L \bmod 2$,

3. $\forall x \in \mathbb{R}, |P(x)|^2 + |Q(x)|^2 = 1$.

Lemma 1 decomposes the unitary matrix of the QNN $U_{\theta,L}^{YZY}(x)$ into Laurent polynomials with real coefficients, which can be used to represent a partial Fourier series with real coefficients. The proof of Lemma 1 uses a method of mathematical induction that is in the similar spirit of the proof of quantum signal processing [32–35], which is a powerful subroutine in Hamiltonian simulation [4] and quantum singular value transformation [35]. The forward direction is straightforward by the definition of $U_{\theta,L}^{YZY}(x)$ in Eq. (10). The proof of the backward direction is by induction in $L$ where the base case $L = 0$ holds trivially. For $L > 0$, we prove that for any $U_{\theta,L}^{YZY}(x)$ where $P, Q$ satisfy the three conditions, there exists a unique block $R_Y^\dagger(\theta_k) R_Z^\dagger(x)$ such that polynomials $\hat{P}$ and $\hat{Q}$ in $U_{\theta,L}^{YZY}(x) R_Y^\dagger(\theta_k) R_Z^\dagger(x)$ satisfy the three conditions for $L - 1$. Lemma 1 explicitly correlates the frequency spectrum of the Fourier series and the number of layers $L$ of the QNN. The proof of Lemma 1 also illustrates the exact correspondence between the Fourier coefficients and parameters of trainable gates. A detailed proof can be found in Appendix A.1.

Other than characterizing the QNN with Laurent polynomials, we also need to specify the achievable Laurent polynomials $P(x)$ for which there exists a corresponding $Q(x)$ satisfying the three conditions in Lemma 1. It has been proved in Refs. [32, 34] that the only constraint is $|P(x)| \le 1$ for all $x \in \mathbb{R}$. That is, for any $P \in \mathbb{R}[e^{ix/2}, e^{-ix/2}]$ with $\deg(P) \le L$ and parity $L \bmod 2$, if $|P(x)| \le 1$ for all $x \in \mathbb{R}$, there exists a $Q \in \mathbb{R}[e^{ix/2}, e^{-ix/2}]$ with $\deg(P) \le L$ and parity $L \bmod 2$ such that $|P(x)|^2 + |Q(x)|^2 = 1$ for all $x \in \mathbb{R}$.

By Lemma 1, the partial Fourier series corresponding to the QNN $U_{\theta,L}^{YZY}(x)$ only has real coefficients. With the exponential form of Eq. (6), a Fourier series with real coefficients only has $\cos(nx)$ terms, which means $U_{\theta,L}^{YZY}(x)$ can be used to approximate any even function on the interval $[-\pi, \pi]$. Thus we establish the following proposition, whose proof is deferred to Appendix A.2.

**Proposition 2** *For any even square-integrable function $f : [-\pi, \pi] \to \mathbb{R}$ and for all $\epsilon > 0$, there exists a QNN $U_{\theta,L}^{YZY}(x)$ such that $|\psi(x)\rangle = U_{\theta,L}^{YZY}(x) |0\rangle$ satisfies*

$$\| \langle \psi(x)|Z|\psi(x)\rangle - \alpha f(x) \| \le \epsilon \tag{12}$$

*for some normalizing constant $\alpha$.*

Although the above result states that the QNN $U_{\theta,L}^{YZY}(x) |0\rangle$ is able to approximate a class of even functions within arbitrary precision, we can see the main limitation of the expressive power of QNN $U_{\theta,L}^{YZY}(x)$ is the real Fourier coefficients, which may restrict its universal approximation capability.

To tackle this issue, our idea is to introduce complex coefficients to the corresponding Laurent polynomials, which can be implemented by adding a trainable Pauli $Z$ rotation operator in each layer. Specifically, we consider the QNN

$$U_{\boldsymbol{\theta},\boldsymbol{\phi},L}^{WZW}(x) = R_Z(\varphi)W(\theta_0,\phi_0)\prod_{j=1}^{L} R_Z(x)W(\theta_j,\phi_j), \tag{13}$$

where each trainable block is $W(\theta_j,\phi_j) \coloneqq R_Y(\theta_j)R_Z(\phi_j)$. Here we add an extra $R_Z(\varphi)$ gate to adjust the relative phase between $P$ and $Q$. The quantum circuit of $U_{\boldsymbol{\theta},\boldsymbol{\phi},L}^{WZW}(x)$ is illustrated in Fig. 3.

Figure 3: Circuit of $U_{\boldsymbol{\theta},\boldsymbol{\phi},L}^{WZW}(x)$, where the trainable block is composed of $R_Y(\cdot)$ and $R_Z(\cdot)$, and the encoding block is $R_Z(\cdot)$.

To characterize the capability of this QNN, we establish the following Lemma which implies $U_{\boldsymbol{\theta},\boldsymbol{\phi},L}^{WZW}(x)$ can express any Fourier partial sum with complex Fourier coefficients.

**Lemma 3** *There exist* $\boldsymbol{\theta} = (\theta_0,\theta_1,\ldots,\theta_L) \in \mathbb{R}^{L+1}$ *and* $\boldsymbol{\phi} = (\varphi,\phi_0,\phi_1,\ldots,\phi_L) \in \mathbb{R}^{L+2}$ *such that*

$$U_{\boldsymbol{\theta},\boldsymbol{\phi},L}^{WZW}(x) = \begin{bmatrix} P(x) & -Q(x) \\ Q^*(x) & P^*(x) \end{bmatrix} \tag{14}$$

*if and only if Laurent polynomials* $P, Q \in \mathbb{C}[e^{ix/2}, e^{-ix/2}]$ *satisfy*

1. $\deg(P) \le L$ *and* $\deg(Q) \le L$,

2. $P$ *and* $Q$ *have parity* $L \bmod 2$,

3. $\forall x \in \mathbb{R}, |P(x)|^2 + |Q(x)|^2 = 1$.

Lemma 3 demonstrates a decomposition of the QNN $U_{\boldsymbol{\theta},\boldsymbol{\phi},L}^{WZW}(x)$ into Laurent polynomials with complex coefficients, which can be used to represent a partial Fourier series with complex coefficients in form of Eq. (8). The proof of Lemma 3 is similar to the proof of Lemma 1 and its details are provided in Appendix A.3. Again, the proof demonstrates the effect of parameterized gates on the control of Fourier coefficients. Similarly, the constraint for the achievable complex Laurent polynomials $P(x)$ in $U_{\boldsymbol{\theta},\boldsymbol{\phi},L}^{WZW}(x)$ is that $|P(x)| \le 1$ for all $x \in \mathbb{R}$, as proved in Refs. [36, 37].

We then prove in the following Theorem 4 that $U_{\boldsymbol{\theta},\boldsymbol{\phi},L}^{WZW}(x)$ is able to approximate any square-integrable function within arbitrary precision, using the well-established result in Fourier analysis. The detailed proof is deferred to Appendix A.4.

**Theorem 4 (Univariate approximation properties of single-qubit QNNs.)** *For any univariate square-integrable function* $f : [-\pi, \pi] \to \mathbb{R}$ *and for all* $\epsilon > 0$, *there exists a QNN* $U_{\boldsymbol{\theta},\boldsymbol{\phi},L}^{WZW}(x)$ *such that* $|\psi(x)\rangle = U_{\boldsymbol{\theta},\boldsymbol{\phi},L}^{WZW}(x)|0\rangle$ *satisfies*

$$\| \langle\psi(x)|Z|\psi(x)\rangle - \alpha f(x) \| \le \epsilon \tag{15}$$

*for some normalizing constant* $\alpha$.

Up till now we only let the encoding gate be the $R_Z(x)$ gate, what if we use other rotation operator gates to encode the data? It actually does not matter which one we choose as the encoding gate if the trainable gates are universal. Note that Pauli rotation operators $R_X(x), R_Y(x), R_Z(x)$ have two eigenvalues $\cos(x/2) \pm i\sin(x/2)$, and they can be diagonalized as $Q^\dagger R_Z(x)Q$. Merging unitaries $Q^\dagger$ and $Q$ to universal trainable gates gives the QNN that uses $R_Z(x)$ as the encoding gate. We hereby define the generic single-qubit QNNs as

$$U_{\boldsymbol{\theta},\boldsymbol{\phi},\boldsymbol{\lambda},L}^{UZU}(x) = U_3(\theta_0,\phi_0,\lambda_0)\prod_{j=1}^{L} R_Z(x)U_3(\theta_j,\phi_j,\lambda_j), \tag{16}$$

where each trainable block is the generic rotation gate

$$U_3(\theta, \phi, \lambda) = \begin{bmatrix} \cos\frac{\theta}{2} & -e^{i\lambda}\sin\frac{\theta}{2} \\ e^{i\phi}\sin\frac{\theta}{2} & e^{i(\phi+\lambda)}\cos\frac{\theta}{2} \end{bmatrix}. \tag{17}$$

By definition, any $L$-layer single-qubit QNN, including $U_{\boldsymbol{\theta},\boldsymbol{\phi},L}^{WZW}$, can be expressed as $U_{\boldsymbol{\theta},\boldsymbol{\phi},\boldsymbol{\lambda},L}^{UZU}$. Thus $U_{\boldsymbol{\theta},\boldsymbol{\phi},\boldsymbol{\lambda},L}^{UZU}$ is surely a universal approximator.

## 4  Limitations of single-qubit QNNs

We have proved that a single-qubit QNN is a universal approximator for univariate functions, it is natural to consider its limitations. Is there a single-qubit QNN that can approximate arbitrary multivariate functions? We answer this question from the perspective of multivariate Fourier series. Specifically, we consider the generic form of single-qubit QNNs defined in Eq. (16) and upload the classical data $\boldsymbol{x} := (x^{(1)}, x^{(2)}, \cdots, x^{(d)}) \in \mathbb{R}^d$ as

$$U_{\boldsymbol{\theta},L}(\boldsymbol{x}) = U_3(\theta_0, \phi_0, \lambda_0)\prod_{j=1}^{L} R_Z(x_j)U_3(\theta_j, \phi_j, \lambda_j), \tag{18}$$

where each $x_j \in \boldsymbol{x}$ and $L \in \mathbb{N}^+$. Without loss of generality, assume that each dimension $x^{(i)}$ is uploaded the same number of times, denoted by $K$. Naturally, we have $Kd = L$. Further, we rewrite the output of QNNs defined in Eq. (5) as the following form.

$$f_{\boldsymbol{\theta},L}(\boldsymbol{x}) = \sum_{\boldsymbol{\omega}\in\Omega} c_{\boldsymbol{\omega}} e^{i\boldsymbol{\omega}\cdot\boldsymbol{x}}, \tag{19}$$

where $\Omega = \{-K, \cdots, 0, \cdots, K\}^d$, and the $c_{\boldsymbol{\omega}}$ is determined by parameters $\boldsymbol{\theta}$ and the observable $M$. A detailed analysis can be found in Appendix B. We can see that Eq. (19) cannot be represented as a $K$-truncated multivariate Fourier series. Specifically, by the curse of dimensionality, it requires exponentially many terms in $d$ to approximate a function in $d$ dimensions. However, for $f_{\boldsymbol{\theta},L}(\boldsymbol{x})$, the degrees of freedom grow linearly with the number of layers $L$. It implies that single-qubit native QNNs potentially lack the capability to universally approximate arbitrary multivariate functions from the perspective of the Fourier series.

Despite the potential limitation of native QNNs in multivariate approximation, it has been proved that a single-qubit **hybrid** QNN can approximate arbitrary multivariate functions [28, 29]. However, the UAP of hybrid QNNs is fundamentally different from the native model that we investigated. Those hybrid models involve trainable weights either in data pre-processing or post-processing. Specifically, introducing trainable weights in data pre-processing is equivalent to multiplying each frequency of the Fourier series by an arbitrary real coefficient, i.e.

$$S(wx) = R_Z(wx) = e^{-iw\frac{x}{2}Z}. \tag{20}$$

Therefore it enriches the frequency spectrum of native QNNs, which only contain integer multiples of the fundamental frequency. It can also be readily extended to the encoding of multi-dimensional data $\boldsymbol{x} := (x^{(1)}, x^{(2)}, \cdots, x^{(d)})$ as

$$R_Z(w_1 x^{(1)})R_Z(w_2 x^{(2)})\cdots R_Z(w_d x^{(d)}) = R_Z(\boldsymbol{w}\cdot\boldsymbol{x}) = e^{-\frac{1}{2}i\boldsymbol{w}\cdot\boldsymbol{x}Z}, \tag{21}$$

where $\boldsymbol{w} = (w_1, \ldots, w_d)$ is a vector of trainable weights. Using such an encoding method enables a single-qubit QNN to approximate any continuous multivariate function [29]. We notice that, although the trainable weights enrich the frequency spectrum of the Fourier series, the capability of hybrid QNNs to approximate arbitrary multivariate functions is not obtained using the multivariate Fourier series, but the universal approximation theorem [25, 26] in machine learning theory. In another word, the multivariate UAP of a hybrid QNN mostly comes from the classical structure, and the QNN serves as an activation function $\sigma(x) = e^{-ix}$ in the universal approximation theorem. This fact might be able to shed some light on the reason why a hybrid QNN does not provide quantum advantages over the classical NN.

# 5  Numerical experiments

In order to better illustrate the expressive power of single-qubit native QNNs, we supplement the theoretical results with numerical experiments. Specifically, we demonstrate the flexibility and approximation capability of single-qubit native QNNs in Section 5.1. The limitations of single-qubit QNNs are illustrated in Section 5.2 through the numerical experiments on approximating multivariate functions. All simulations are carried out with the Paddle Quantum toolkit on the PaddlePaddle Deep Learning Platform, using a desktop with an 8-core i7 CPU and 32GB RAM.

## 5.1  Univariate function approximation

A damping function $f(x) = \sin(5x)/5x$ is used to demonstrate the approximation performance of single-qubit native QNN models. The dataset consists of 300 data points uniformly sampled from the interval $[0, \pi]$, from which 200 are selected for the training set and 100 for the test set. Since the function $f(x)$ is an even function, we use the QNN model as defined in Eq. (10). The parameters of trainable gates are initialized from the uniform distribution on $[0, 2\pi]$. We adopt a variational quantum algorithm, where a gradient-based optimizer is used to search and update parameters in the QNN. The mean squared error (MSE) serves as the loss function. Here the Adam optimizer is used with a learning rate of 0.1. We set the training iterations to be 100 with a batch size of 20 for all experiments.

While approximating a function $f(x)$ by a truncated Fourier series, the approximation error decreases as the number of expansion terms increases. As shown in Lemma 3, the frequency spectrum and Fourier coefficients will be extended by consecutive repetitions of the encoding gate and trainable gate. The numerical results in Fig. 4 illustrate that the approximation error decreases as the number of layers increases, which are consistent with our theoretical analysis.

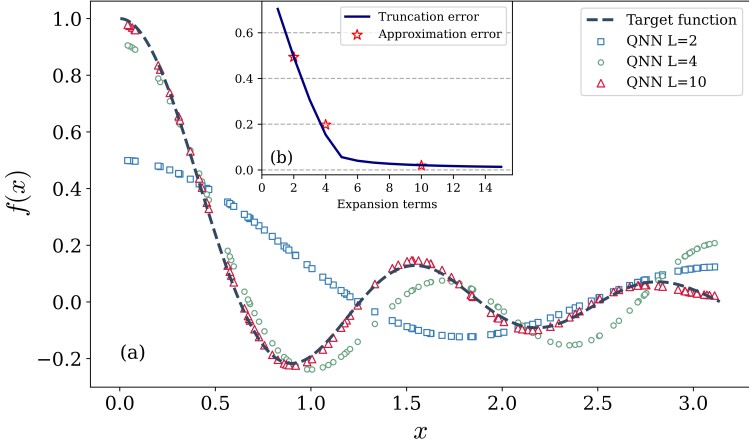

Figure 4: Panel (a) is the approximation results of $U_{\boldsymbol{\theta},L}^{YZY}$ with different $L$. Panel (b) shows the approximation error. Here the theoretical truncation error is calculated using the uniform norm $\|f_N(x) - f(x)\|_\infty$, where $f_N(x)$ is the truncated Fourier series of $f(x)$ with $N$ terms.

To further show the flexibility and capability of single-qubit QNNs, we pick a square wave function as the target function. The training set contains 400 data points sampled from the interval $[0, 20]$. The numerical results are illustrated in Fig. 5. By simply repeating 45 layers, the single-qubit QNN $U_{\boldsymbol{\theta},\boldsymbol{\phi},L}^{WZW}(x)$ learns the function hidden beneath the training data. In particular, the approximation works well not only for input variables located between the training data but also outside of the region, because the Fourier series has a natural capability in dealing with periodic functions.

## 5.2  Multivariate function approximation

We numerically demonstrate the limitations of single-qubit native QNNs in approximate multivariate functions. We examine the convergence of the loss as the number of layers of the circuit increases and compare the outcome with the target function. Specifically, we consider a bivariate function

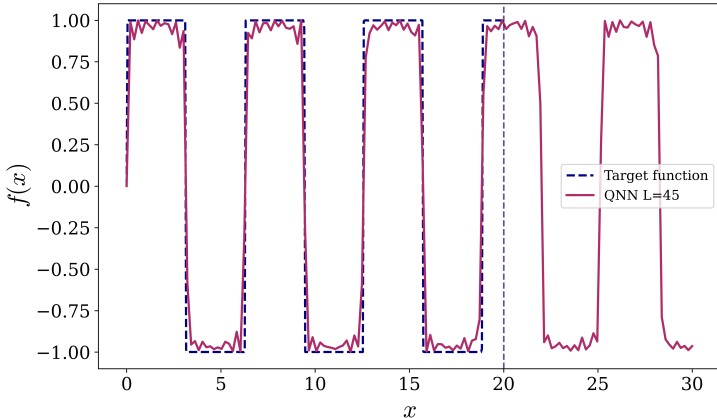

Figure 5: The result of approximating the square wave function by a $45$-layer QNN $U_{\boldsymbol{\theta},\boldsymbol{\phi},L}^{WZW}(x)$. For showing periodic extrapolation, and the test region is extend to $[0,30]$.

$f(x,y) = (x^2 + y - 1.5\pi)^2 + (x + y^2 - \pi)^2$ as the target function. Note that $f(x,y)$ is normalized on the interval $[-\pi,\pi]^2$, i.e., $-1 \leq f(x,y) \leq 1$.

The training set consists of 400 data points sampled from interval $[-\pi,\pi]^2$. We use the single-qubit QNN with various numbers of layers defined as Eq. (18) to learn the target function. The experimental setting is the same as in the univariate function approximation. In order to reduce the effect of randomness, the experimental results are averaged over 5 independent training instances.

Fig. 6 shows that the single-qubit native QNN has difficulty in approximating bivariate functions. The approximation result of QNN as shown in Fig. 6b is quite different from the target function, even for a very deep circuit of 40 layers. Also, the training loss in Fig. 6c does not decrease as the number of layers increases. Note that the target function is only bivariate here, the limitations of single-qubit native QNNs will be more obvious in the case of higher dimensions. We further propose a possible strategy that extends single-qubit QNNs to multiple qubits, which could potentially overcome the limitations and handle practical classification tasks, see Appendix C for details.

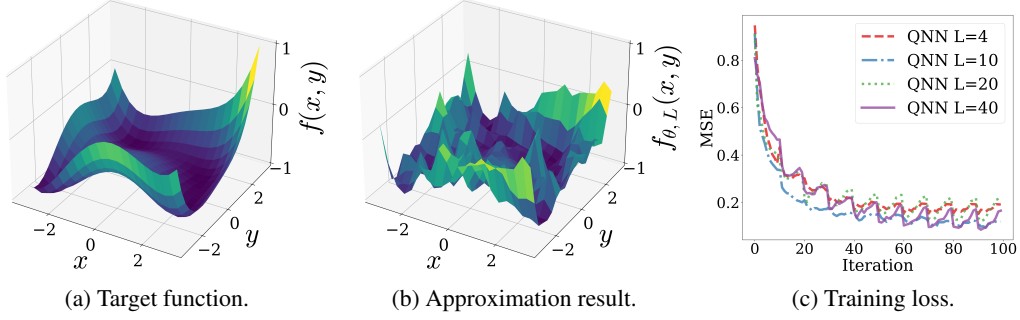

(a) Target function.  (b) Approximation result.  (c) Training loss.

Figure 6: Panel (a) is the plot of target function $f(x,y)$. Panel (b) shows the approximation result of a $40$-layer QNN. Panel (c) is the plot of training loss for QNNs with different numbers of layers during the optimization.

## 6 Conclusion and outlook

In this work, we presented a systematic investigation of the expressive power of single-qubit native QNNs, which are capable to approximate any square-integrable univariate function with arbitrary precision. We not only give an existence proof but also analytically show an exact mapping between native QNNs and the partial Fourier series from perspectives of both frequency spectrum and Fourier coefficients, which solves an open problem on the UAP of single-qubit QNNs in Ref. [27]. Our proof, inspired by quantum signal processing, explicitly illustrates the correlation between parameters of

trainable gates and the Fourier coefficients. Other than the expressivity, we also discuss the limitation of single-qubit QNNs from the perspective of multivariate Fourier series. Both the expressivity and limitation of single-qubit QNNs are validated by numerical simulations. We expect our results provide a fundamental framework to the class of data re-uploading QNNs, which serves as insightful guidance on the design of such QNN models.

Although the expressive power of a single-qubit QNN have been well investigated, it may not be an ideal model in practice due to the potential limitations on approximating multivariate functions. Moreover, single-qubit models can be efficiently simulated by classical computers and hence cannot bring any quantum advantage. The multi-qubit QNNs as shown in Ref. [27] and in Appendix C might require exponential circuit depth, which is impractical to implement and also does not fit the systematic analysis for the single-qubit case. Therefore one future step is to efficiently generalize the framework of single-qubit QNNs to multi-qubit cases. One promising approach is to encode data into multi-qubit unitaries by block encoding and then mapping higher-dimensional operations on multi-qubit systems to single-qubit gates by qubitization [38]. Such techniques are originally used in multi-qubit extensions of quantum signal processing, such as quantum singular value transformation [35] and quantum phase processing [37]. By the connection between single-qubit QNNs and quantum signal processing, block encoding and qubitization may lead to useful QNN models for multi-qubit cases and establish corresponding systematic analyses. A recent paper presents a method that extends quantum signal processing to multivariate [39], which might also be applicable to single-qubit QNNs. We believe our results and their possible extensions would improve our understanding of QNNs and provide a helpful guideline for designing powerful QNNs for machine learning tasks.

## Acknowledgments and Disclosure of Funding

We would like to thank Runyao Duan for helpful suggestions on quantum signal processing. We also thank Guangxi Li, Geng Liu, Youle Wang, Haokai Zhang, Lei Zhang, and Chengkai Zhu for useful comments. Z. Y. and H. Y. contributed equally to this work. Part of this work was done when Z. Y., H. Y., and M. L. were research interns at Baidu Research.

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
