# OpenReview forum: "Power and limitations of single-qubit native quantum neural networks"
_NeurIPS.cc/2022/Conference — NeurIPS 2022 Accept_

### Official Review · Reviewer_tdtX · 2022-07-11

**Rating:** 5
**Confidence:** 2
**Soundness:** 3 good
**Presentation:** 2 fair
**Contribution:** 2 fair

**Summary:**

The authors show that a data reuploading QNN with only $R_Z(x)$ encoding gates and $R_Y(\cdot)$ as trainable gates can be represented as a truncated Fourier series with real coefficients. Thus QNNs of this form can approximate even functions arbitrarily well. The authors extend the QNN model with trainable gates of the from $W(\theta_j, \phi_j):= R_Y(\theta_j) R_Z(\phi_j)$ to represent complex Fourier coefficients. Using established results from Fourier analysis the authors can show that the extended QNN model is a universal function approximator in the class of square-integrable functions. Within the Fourie series framework the authors provides arguments for the inadequacy of their QNN ansatz to universally approximate multivariate functions. Numerical experiments were performed to show the increasing approximation capability with increasing number of layers of the QNN model and also the extrapolation ability of the QNN was demonstrated for a simple periodic function. In the end the single qubit QNN is tested on a bivariate function, where the performance is as expected not optimal.


**Questions:**

- What is the difference between Lemma 1 and 3 except for the additionals parameters?
- Why do you use only 200+100 datapoints for training/testing of your 1000 point dataset?
- Why do you use the damping function? Have you considered a more diverse set of functions?
- will the code be available to reproduce the findings?


**Limitations:**

The authors adequately show the limitation of their model in the bivariate case. No societal impact was discussed.

**Strengths And Weaknesses:**

Pro
- the foundational presentation of the topic was clearly structured

Cons:
- the universal approximations theorem for QNNs was already proven in M.Schuld et 2021
- the concept of data reuploading was also already established in A.Perez-Salinas et al 2020
- thus the originality of the paper is not clear
- though the prior work was cited, the results were not discussed and were not set into perspective with this work

---

> ### Author Response · Authors · 2022-08-02
> **Answer To Reviewer tdtX**
>
> We thank the reviewer for their time and their comments. We would like to address your critical assessment that the originality of the paper is not clear.
>
> >Con1: The concept of data reuploading was already established in Perez-Salinas et al 2020.
>
> Re1: The powerful data re-uploading QNN model was first established in Perez-Salinas et al 2020. The authors also numerically showed the potential of data re-uploading QNN model as a quantum classifier. However, we want to emphasize that the theoretical analysis was not established in this paper, which limits the interpretability of QNN models. We will elaborate more in Re3 about our contributions on how we overcome this difficulty.
>
> >Con2: The universal approximations theorem for QNNs was already proven in Schuld et al 2021.
>
> Re2:
> The paper by Schuld et al. proved the UAT of native QNN models, which showed the powerful expressivity of QNN models. But we want to point out that the existence proof in this work is under the strong assumption of multi-qubit systems, exponential circuit depth, and arbitrary observables, which does not explicitly give the structure of QNNs. The UAT for single-qubit models was left as a key open problem for fully understanding this powerful model (You can also refer to [Quantum models as Fourier series](https://pennylane.ai/qml/demos/tutorial_expressivity_fourier_series.html)). We will elaborate more in Re3 about our contributions on how we solve the open problem and address the technical limitations.
>
> >Con3: The originality of the paper is not clear.
>
> Re3: We strongly believe that our work presents technically profound results and original contributions which we elaborate next.
> - The previous existence proof on universality did not explicitly give the structure of QNNs, which hinders the understanding of how the structural properties of QNNs may affect their expressive power. We overcome this critical challenge by giving constructive proofs that establish an exact correspondence between the truncated Fourier series and the structure of single-qubit QNNs. This clearly solves an open problem on the universal approximation property of single-qubit QNN models left in Schuld et al 2021. Based on our single-qubit analysis, we have more chances to extend the QNN to an efficient multi-qubit model, which advances and guides the design of QNN.
> - We systematically investigate how trainable circuit blocks affect the Fourier coefficients, which gives a constructive analysis on the UAT of QNN beyond the proof of existence in Schuld et al 2021. This is the first quantitative and rigorous analysis of the relationship between the Fourier coefficients and the trainable parameters of single-qubit QNNs, which significantly improves the interpretability of QNNs.
>
> >Con4: Though the prior work was cited, the results were not discussed and were not set into perspective with this work.
>
> Re4: Indeed, we clearly discuss the results of the previous works and the originality and contributions of our work in the Introduction. In Preliminaries, we give a definition of QNNs proposed by Perez-Salinas et al. We also emphasize the difference between the native and hybrid QNNs. In Section 4, we analyze hybrid single-qubit QNNs on multivariate functions approximation.
>
> Questions:
>
> >Q1: What is the difference between Lemma 1 and 3 except for the additional parameters?
>
> A1: Except for the additional parameters, we also modify the structure of QNNs. In Lem 1, we use $R_Y$ as the trainable gate and show that the QNN can only represent Fourier series with real coefficients. In Lem 3, we add extra $R_Z$ trainable gates to make the Fourier coefficients complex numbers.
>
> >Q2: Why do you use only 200+100 data points for training/testing of your 1000-point dataset?
>
> A2: Thank you for pointing it out. There is actually a mistake on writing, the 1000 points are evenly-spaced sampled (not uniformly sampled as written in the paper) on $[0, \pi]$, which are used to draw the target function in the figure. For convenience, we uniformly sample 200+100 data points for training/testing from the 1000 points. We are sorry for the confusion caused by this writing mistake and we have fixed it in the rebuttal revision.
>
> >Q3: Why do you use the damping function? Have you considered a more diverse set of functions?
>
> A3: The purpose of experiments is to verify our theorems, i.e., any square-integrable function can be approximated, which is a very large class of functions. Thus we just select two typical examples to approximate: the damping function is a smooth, non-periodic even function, and the square wave is a non-smooth periodic odd function. These two functions also have important applications in physics and signal processing. We believe our examples are diverse enough and sufficient to verify our theorems.
>
> >Q4: Will the code be available to reproduce the findings?
>
> A4: We provide all the codes in the Supplementary Material, please check it if of interest.

---

> > ### Comment · Reviewer_tdtX · 2022-08-09
> > **Answer to the Authors**
> >
> > I thank the authors for adressing my comments.
> > With those comments and the surrounding discussion on this platform I can now better understand and appreciate the significance of the authors findings and thus I am willing to reassess my initial rating.

---

### Official Review · Reviewer_YkGj · 2022-07-11

**Rating:** 4
**Confidence:** 3
**Soundness:** 3 good
**Presentation:** 3 good
**Contribution:** 2 fair

**Summary:**

The paper proves that single-qubit quantum neural networks can approximate any univariate function. Both the expressivity and limitation of single-qubit QNNs are validated by numerical simulations involving also multivariate functions.

**Questions:**

What would happen for the multi qubit case? Can you establish a representation theorem there?

**Limitations:**

yes

**Strengths And Weaknesses:**

Strength: a controlled setup to study the approximation capabilities of a single-qubit QNN

Weakness: the setup seems really too simple to capture any of the complexity of representing multi variate functions and, especially, the benefits of using entangling gates. This single qubit analysis is not really illuminating in this sense. The power of the universal approximation theorem in the classical case is that it can be extended to prove an analogous result for multi variate functions, instead in this case it is not clear how and if the proof carried in this work can be used in the more general (and interesting) case.

---

> ### Author Response · Authors · 2022-08-02
> **Answer To Reviewer YkGj**
>
> We thank the reviewer for their time and their valuable comments.  We would like to provide a detailed response to the questions raised by the reviewer.
>
> >Comments: The setup seems really too simple to capture any of the complexity of representing multi-variate functions and, especially, the benefits of using entangling gates. This single qubit analysis is not really illuminating in this sense. The power of the universal approximation theorem in the classical case is that it can be extended to prove an analogous result for multi-variate functions, instead in this case it is not clear how and if the proof carried in this work can be used in the more general (and interesting) case.
>
> Re: We would like to emphasize that systematically analyzing the expressivity of single-qubit native QNNs is quite a non-trivial task. First, establishing an exact mapping between QNN models and Fourier series without any assumption is technically difficult. Before our work, the expressivity analysis of QNNs is under strong assumptions of multi-qubit systems, exponential circuit depth, and arbitrary observables, which yields an existence proof rather than a constructive proof. The universality of single-qubit QNNs was in particular left as an open problem in Schuld et al. 2021. Second, systematically analyzing the relationship between the Fourier coefficients and the trainable parameters in single-qubit gates is quite challenging since the interleaved structure of data re-uploading QNN, the Fourier coefficients and the trainable parameters are not trivially one-to-one mapped. Before our work, these unsolved problems hindered the understanding of the expressivity and limitation of QNN, especially on how structural properties of QNN may affect its expressive power. A lack of constructive analysis makes it highly challenging to design effective and efficient QNNs.
>
> Our work, notably, constructively establishes an expressivity analysis of the single-qubit QNN models beyond previous existence proofs and systematically investigates how trainable parameters affect the Fourier coefficients. This solid and significant contribution exactly solves the challenging problems mentioned above. Now with our work, we can clearly interpret the relationship between the structural properties of QNN and its expressive power, which serves as insightful guidance on the design of QNN models for future work. For example, the multi-qubit extension that we proposed is an experimental attempt inspired by our single-qubit analysis, which shows potential for approximating multivariate functions. Extending the universality to multi-variate functions approximation in the quantum scenario is totally different from the classical case since the number of parameters in multi-qubit universal quantum gates grows exponentially as the number of qubits, which makes a systematic analysis extremely difficult. Indeed, our theoretical proofs are hard to be generalized directly to multi-qubit cases that we proposed, but they could be potentially generalized to higher dimensions by using some quantum techniques like qubitization, see our answer to the question below for more details. Overall, the generalization and theoretical analysis of multi-qubit models are out of the scope of our paper, and we do not think lacking such analyses is a critical weakness of our paper.
>
> >Q: What would happen for the multi-qubit case? Can you establish a representation theorem there?
>
> A: We really appreciate this insightful question that motivates us to think further about the problem of multi-qubit extension. The multi-qubit model that we proposed in Appendix C of the paper is not the only way to extend it, there are also some other ways to extend the model to a higher dimension. For example, there is a technique in quantum computing called ``qubitization'' ([Hamiltonian Simulation by Qubitization](https://arxiv.org/abs/1610.06546)) that embeds (qubitizes) higher-dimensional phase operations to single-qubit phase gates. In other words, one can use a sequence of single-qubit gates to control a multi-qubit system by transforming the singular values of the Hamiltonian, i.e., to extend our univariate function $f(x)$ to a function $f(H)$, where $H$ is the higher-dimensional Hamiltonian.
> Thus, using this technique, one will have the chance to introduce useful QNN models for the multi-qubit case and establish corresponding systematic analyses.
>
> Overall, we would like to clarify that it is possible to extend our theoretical analysis to multiple qubits via some recently developed quantum techniques. Such extension using qubitization will centrally rely on the rigorous theory of single-qubit QNN, which is exactly the main focus of our work. The extension is somehow out of the scope of this paper, but definitely interesting for future work. Thank you for your inspiring question again and we will elaborate more on this perspective in the revision.

---

> > ### Comment · Reviewer_YkGj · 2022-08-08
> > **Existence and multi-qubit**
> >
> > I thank the authors for their reply to my comments. I now better appreciate the technical importance of the result, especially in light of the fact that it is a constructive one rather than a mere existence result (already existing in literature).
> >
> > While I believe that the impact of this work is still somehow limited, because it does not address the multi-qubit case, I am willing to reconsider my initial assessment and consider the work suitable for acceptance.

---

### Official Review · Reviewer_Npzq · 2022-07-11

**Rating:** 7
**Confidence:** 4
**Soundness:** 4 excellent
**Presentation:** 4 excellent
**Contribution:** 3 good

**Summary:**

This paper aims to explore the data re-uploading expressive ability of quantum neural networks (QNNs) by providing rigorous theoretical proofs for native single-qubit QNNs. They point out that native QNNs demonstrate the expressivity of the quantum part of hybrid QNNs, decoupled from the classical part of hybrid QNNs. In a series of proofs, they show that single-qubit native QNNs can approximate any univariate square-integrable function arbitrarily close by an exact mapping to the partial Fourier series. They then discuss how the native QNNs lack the expressivity in approximating multivariate functions and hybrid single qubit QNNs can approximate arbitrary multivariate functions with the help of classical structures. They further show that increasing the number of QNN laybers decreases the mean squared error for approximating a function.

**Questions:**

It is mentioned that hybrid QNNs provide no quantum advantage over classical neural networks. So do the results of this paper prove, or show a potential of quantum advantage for native QNNs? If not, will a multivariate function approximating native QNN provide quantum advantage?

**Ethics Review Area:**

["I don’t know"]

**Limitations:**

The authors have adequately addressed the limitations of native single-qubit QNNs as a trivial case for data loading into QNNs using a partial fourier series.

**Strengths And Weaknesses:**

The paper provides rigorous proofs that build up the foundational work for native single-qubit QNNs in a purely quantum context. It presents, in a logical and consistent manner, how univariate square-integrable functions can be loaded into single-qubit QNNs up to arbitrary precision using the partial Fourier series. The numerical experiments show clear evidence for the correlation between layer number and loss convergence, and the inclusion of the square wave is excellent as it requires a large number of fourier terms to approximate. It also does a good job explaining with theory and demonstrating with an experiment how multivariate functions can not be approximated well with single qubit QNNs.

The paper talks about the limitations of single-qubit QNNs and the importance of “investigating QNNs with universal approximation properties for multivariate functions”. It also provides a potential design for multi-qubit QNNs in Appendix C. However, it is not immediately clear how the results of this paper, especially theoretical proofs which are exclusively on single-qubit QNNs, can be generalized for future studies of multivariate function approximation in QNNs.

---

> ### Author Response · Authors · 2022-08-02
> **Answer To Reviewer Npzq**
>
> We thank the reviewer for the appreciation of the paper and inspiring feedback. Here we respond to the insightful comments and questions.
>
> > Comment: It is not immediately clear how the results of this paper, especially theoretical proofs which are exclusively on single-qubit QNNs, can be generalized for future studies of multivariate function approximation in QNNs.
>
> Re: Indeed, our theoretical proofs are difficult to be directly generalized to the multi-qubit cases that we proposed, but they could be potentially generalized to higher dimensions by using some quantum computing techniques like qubitization. Please refer to our reply to Reviewer (YkGj) for more details.
>
> > Q: Do the results of this paper prove, or show a potential quantum advantage for native QNNs? If not, will a multivariate function approximating native QNN provide a quantum advantage?
>
> A: Thanks for asking this insightful question about quantum advantage. Demonstrating a potential quantum advantage is one of the most urgent goals in the field of quantum machine learning. Our results, which mainly focus on expressivity, do not show the quantum advantage of single-qubit QNNs. As the number of qubits increases, it would be a lot more difficult for a classical computer to efficiently simulate a multi-qubit quantum system. Thus we believe that multi-qubit native QNN models or their high-dimensional generalizations will have potential quantum advantages, especially when combined with quantum algorithms like Hamiltonian simulation. We suggest referring to our reply to the Reviewer (YkGj) for more details on this perspective.

---

> > ### Comment · Reviewer_Npzq · 2022-08-09
> > **keep the score**
> >
> > I thank the authors for the responses which answer my questions to some extents. I would like to keep my score.

---

### Official Review · Reviewer_LZ4h · 2022-07-26

**Rating:** 6
**Confidence:** 3
**Soundness:** 3 good
**Presentation:** 3 good
**Contribution:** 3 good

**Summary:**

In this paper the authors analyze the single qubit neural networks by using the re-uploading QNN formalism. The contributions of the paper are the proofs that a single qubit neural network, simply a single qubit quantum circuits with half of parameterizable gates can approximate any single bit uni-variate function. In additions the authors provide analysis of the re-uploading networks for simulating multi-variate functions. The authors show that the re-uploading networks are not universal in approximating multi-variate functions.

**Questions:**

What is a reasonable approximation of multi-variate function as to the dimension d in Omega and error e?

**Limitations:**

No ethical concerns have been determined

**Strengths And Weaknesses:**

The paper analysis of single qubit networks in the re-uploading format, limited by the Fourier-transform blocks is a nice extension to the original re-uploading NN.The proof of the univariate function approximation is a simpler form of the universal circuit Solovay-Kitaev  approximation theorem. The principle of the proof starting from eq.4 can be simplified by assuming that if the target function is expressed simply by V(\theta_0) then the sum over S(x)V(\theta_j) = I. That is each of the encoding block can be nilled by finding exactly the inverse transform by learning. For the multi-variate approximation, it is true that the studied setting is not universal.

While both of the studied problems are correctly analyzed I do wonder if this is really related to QNN. In particular, now when most of the algorithms are being built as VQE I wonder if this is the correct venue.

---

> ### Author Response · Authors · 2022-08-02
> **Answer To Reviewer LZ4h**
>
> We thank the reviewer for acknowledging that our paper is technically sound and providing a possible way to simplify the proofs.  Here we respond to the insightful comments and questions.
>
> > Comment 1: Analysis related to QNN?
>
> Re1: Thanks for pointing out the potential relationship between our results and the Solovay–Kitaev theorem, although they are slightly different. We would like to clarify that our theorems analyze the ability of single-qubit gates (in a data re-uploading form) to exactly represent the truncated Fourier series and then approximate a function of data, while the Solovay–Kitaev theorem describes the ability of universal gates to approximate a unitary. QNN, just like the classical NN, is a function model that maps inputs to outputs. Thus our analysis is closely related to QNN in the context of ``function'', with a particular focus on the expressivity of QNN.
>
> > Comment 2: Is this the correct venue? In particular, now most of the algorithms are being built as VQE.
>
> Re2: Thank you for this question. We believe this work is suitable for this venue. First, hybrid quantum-classical or variational quantum algorithms (VQA) is a mainstream of quantum machine learning models with wide applications [1, 2, 3]. Second, the data re-uploading QNN considered in our work is a general quantum machine learning model that unifies all standard quantum models based on parameterized quantum circuits (PQC), including kernel methods and linear quantum models [4]. Notably, our work solves the fundamental problem on the theoretical expressivity of single-qubit QNN, which is of broad interest in most directions of quantum machine learning, especially in the interpretability of QNNs. Third, there are many great works based on VQAs and QNNs presented in AI conferences like AAAI and NeurIPS [5, 6, 7, 8, 9], which significantly advanced the research and study of quantum machine learning. Based on the reasons above, we think our work is perfectly suitable for the NeurIPS conference and will be a standard reference for the expressivity and interpretability theory of QNN.
>
> [1] Cerezo, M., et al. “Variational Quantum Algorithms.” Nature Reviews Physics, vol. 3, no. 9, 9, Sept. 2021, pp. 625–44.
>
> [2] Benedetti, Marcello, et al. “Parameterized Quantum Circuits as Machine Learning Models.” Quantum Science and Technology, vol. 4, no. 4, Nov. 2019, p. 043001.
>
> [3] Chen, Samuel Yen-Chi, et al. “Variational Quantum Circuits for Deep Reinforcement Learning.” IEEE Access, vol. 8, 2020, pp. 141007–24.
>
> [4] Jerbi, Sofiene, et al. “Quantum Machine Learning beyond Kernel Methods.” ArXiv:2110.13162 [Quant-Ph, Stat], Feb. 2022.
>
> [5] Li, Guangxi, et al. “VSQL: Variational Shadow Quantum Learning for Classification.” Proceedings of the AAAI Conference on Artificial Intelligence, vol. 35, no. 9, 9, May 2021, pp. 8357–65.
>
> [6] Lockwood, Owen, and Mei Si. “Reinforcement Learning with Quantum Variational Circuit.” Proceedings of the AAAI Conference on Artificial Intelligence and Interactive Digital Entertainment, vol. 16, no. 1, 1, Oct. 2020, pp. 245–51.
>
> [7] Ricks, Bob, and Dan Ventura. “Training a Quantum Neural Network.” Advances in Neural Information Processing Systems, vol. 16, MIT Press, 2003. Neural Information Processing Systems.
>
> [8] Bausch, Johannes. “Recurrent Quantum Neural Networks.” Advances in Neural Information Processing Systems, vol. 33, Curran Associates, Inc., 2020, pp. 1368–79. Neural Information Processing Systems.
>
> [9] Kübler, Jonas, et al. “The Inductive Bias of Quantum Kernels.” Advances in Neural Information Processing Systems, vol. 34, Curran Associates, Inc., 2021, pp. 12661–73. Neural Information Processing Systems.
>
> Questions:
>
> > Q: What is a reasonable approximation of a multi-variate function as to the dimension $d$ in $\Omega$ and error $\epsilon$?
>
> A: Thank you for the insightful question. As we analyzed in Section 4, for the single-qubit QNN to approximate a $d$-variate function, the degree of freedom grows linearly with the number of layers $L=Kd$, while the number of terms in the $d$-variate Fourier series grows approximately exponentially in dimension $d$ by the curse of dimensionality. Because of this huge gap, to the best of our knowledge, there is no non-trivial bound of the approximation error for a single-qubit QNN to approximate an arbitrary $d$-variate function. We will add discussion on this in the revision. As for the multi-qubit cases, we believe there exists a $d$-qubit QNN model that can approximate a $d$-variate function with high precision and we give a toy experimental example in Appendix C, but the theoretical analysis is out of scope for this paper.

---

### Meta-Review · Area_Chair_iyMf · 2022-08-27

**Recommendation:** Accept
**Confidence:** Less certain

**Metareview:**

This submission is borderline.  Reviewers generally agreed that its theoretical contribution is sound and non-trivial, but at the same time pointed to a major limitation,, which is that the theory applies only to single Qubit QNNs, thus disregards the aspect of entanglement, which is one of the most important factors in quantum computational systems.  This point is clearly acknowledged by the text (including the title), so I do not think much more can be done on the authors' part.  I recommend that the paper be accepted, while highly encouraging the authors to at least provide some directions as to how their theory might be relevant to the multi Qubit case.

**Award:**

No

---

### Decision · Program_Chairs · 2022-09-14

Accept